

# Quantity and Quality Benefits of in-Service Invasive Cleaning of Trunk Mains

Iftekhar Sunny[1], Stewart Husband[1], Nick Drake[2], Kevan Mckenzie[2], Joby Boxall[1]

[1]Pennine Water Group, Department of Civil and Structural Engineering, University of Sheffield, S1 3JD, Sheffield (UK)
5 [2]Scottish Water, Juniper House, Heriot Watt Research Park, Edinburgh, EH14 4AP, (UK)

*Correspondence to*: Iftekhar Sunny (izsunny1@sheffield.ac.uk)

**Abstract.** Trunk mains are high risk critical infrastructure where poor performance can impact on large numbers of customers. Both quantity (e.g. hydraulic capacity) and quality (e.g. discolouration) of trunk main performance are affected by asset deterioration in the form of particle accumulation at the pipe wall. Trunk main cleaning techniques are therefore 10 desirable to remove such material. However little is quantified regarding the efficacy of different maintenance interventions, or longer term changes following such cleaning. This paper presents an assessment for quantity and quality performance of a trunk main system pre, post and for twelve months following cleaning using pigging with ice slurry. Hydraulic calibration showed a 7x roughness height reduction after ice slurry pigging, evidencing substantially improved hydraulic capacity and reduced headloss. Turbidity response due to carefully imposed shear stress increase remained significant after the cleaning 15 intervention evidencing that relatively loose materials had be not been fully removed from the pipe wall. Ongoing material accumulation and associated discolouration risk was shown, with the rate of accumulation found to correlate with water temperature. Overall the results demonstrate that cleaning by pigging with ice slurry can be beneficial for quantity performance, but care and further assessment may be necessary to realise the full quality benefits.

## 1 Introduction

20 One of water company's primary responsibilities is to operate and maintain their distribution network performance to ensure the continuous supply of safe, high water quality. As part of the drinking water network, transmission (trunk) mains are categorised as critical infrastructure as poor performances can impact a large numbers of downstream customers. Due to strict operational and quality regulations and concern regarding potential consequences, UK water companies have tended to avoid operational activities associated with trunk mains (Husband and Boxall, 2015). However this is becoming unavoidable 25 and water utilities rehabilitation programs now include large undertaking for the cleaning of large diameter mains (i.e. trunk mains) to manage asset resilience and reduce water quality risks. Considering the massive TOTEX (total expenditure) allocated for cleaning interventions, infrastructure performance assessment both for pre-post cleaning water quality and quantity are vital to justify the investment.



Quantity performances, hydraulic capacity and pipe roughness, impacts the quantity of water received by the consumer, fire flow capacity and energy (pumping) costs. Continuous fouling, tubercles and scaling can increases hydraulic resistance and reduce effective pipe diameter (Boxall et al., 2004). Microbial activity on pipe wall can also raise boundary resistance and affect hydraulic capacity (Cowle et al., 2014).

Discolouration is the water quality issue most apparent to customer, causing the highest contact rates worldwide. The internationally accepted PODDS (Prediction of Discolouration in Distribution Systems) research has verified that discolouration processes are dominated by particulate material (2-25 $\mu$m) continuously attaching to the pipe wall (Boxall and Saul, 2005). These particles are structured in a stable cohesive layers and are released in to the bulk water once an excess shear is imposed. Recent studies have indicated that biofilms are a key component of the discolouration process by

facilitating inorganic particle absorption within the organic matrix (Douterelo et al., 2014).  PODDS concepts are well accepted for smaller diameter pipes in the distribution system, and are started to be shown valid for large diameter transmission mains (Husband and Boxall, 2015).

As the infrastructure ages, particulate material attached to the pipe wall as well as rising biofilm activity can create a significant discolouration risk and reduce mains hydraulic capacity (Cook and Boxall, 2011; Shahzad and James, 2002). To

remove accumulated material, water utilities often use invasive cleaning strategies as part of their rehabilitation programs (AWWA, 2014). It is well established that invasive cleaning (e.g. ice pigging, air scouring, swabbing) can remove significant amounts of accumulated materials and biofilms from operational trunk mains (Friedman et al., 2012). However, little is quantified regarding the efficacy of such interventions in terms of either quantity or quality improvement and how this changes over time following the intervention.

The aim of this paper is to investigate the quantity and quality performance benefits of an in-service trunk main invasive cleaning program. Quantity performance of the selected trunk main was assessed through hydraulic modelling of monitored pre and post invasive cleaning flow and pressure data, with performance improvement evaluated by change in calibrated roughness height. Quality performance was assessed by monitoring, modelling and comparing the turbidity response due to controlled increase in hydraulic conditions pre and post intervention. Post cleaning evaluations where repeated over a

twelve-month period to explore longer term performance changes.

## 2 Methodology

### 2.1 Selected Network Characteristics

The selected trial trunk main was 2.4 km of 228 mm internal diameter asbestos cement (AC) pipe. It provides supply from a service reservoir (SR) into distribution, hence having a demand driven diurnal daily flow pattern with an average

downstream pressure of 42 m and typical daily demand between 6-20 l/s (0.14-0.48 m/s). The supplied DMAs (District Metered Area) were mostly residential with consistent demand across the year. The trunk main operates under gravity, with a low point in the long section as shown in figure 1a.



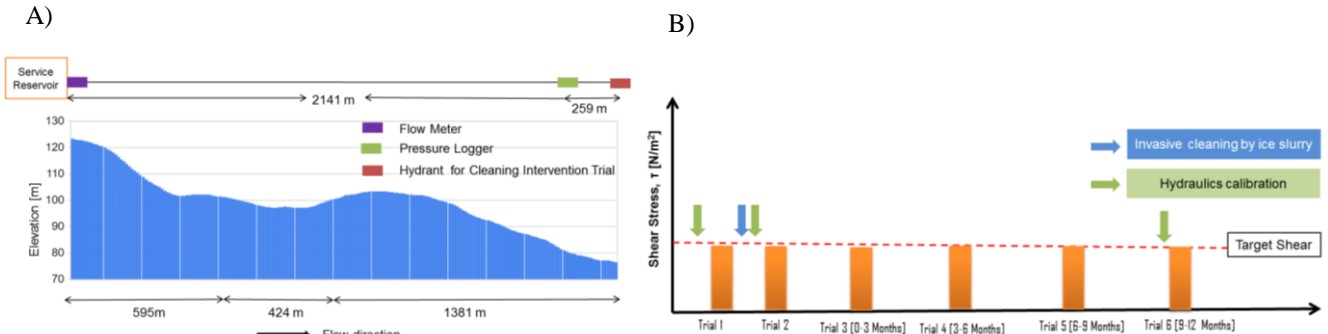

**Figure 1: A) Trunk main layout, monitoring points with elevation profile, B) cleaning interventions planning and timelines**

### 2.2 Fieldwork Procedure and Monitoring

The trunk main was built around 40 years ago. No known cleaning intervention have ever been implemented. As part of a scheme to reduce discolouration customer contacts in the supplied DMAs and to remove particulate material and biofilm attached to the pipe wall, pigging with ice slurry was selected and implemented (Moore, 2013). The method uses ice slurry to form a dynamic plug which increases shear on the pipe wall to dislodge built-up material. A site-specific amount of semi-solid ice slurry was pumped through a UK standard washout hydrant at the trunk main inlet and emitted from hydrant a 2.4 km downstream. Before cleaning, the main was isolated from supply to ensure no mixing of water into supply with the semi solid ice pig.

The trunk main served three separate downstream DMAs, with no connection over the 2.4km length selected for cleaning. This layout ensures constant flow over the entire pipe length. Hence flow at the inlet, service reservoir level data and downstream pressure data allow for accurate estimation of roughness height ($k_s$), quantity performance. Pre and post pressure data was collected using Syrinix Transientminder loggers with 15 minute resolution. Flow was monitored continuously at the service reservoir (SR) outlet using existing instrumentation, again at 15 minute resolution.

Quality performance was assessed by imposing carefully managed flow increases and monitoring for any turbidity response, see figure 1b. From the PODDS concepts the imposed excess (above normal daily peak) shear stress would induced mobilisation of loose material associated with discolouration, with repeated operations of the same magnitude providing insight into any inter-period accumulation or other change in material layers. Managed increases in system shear stress where imposed through a hydrant standpipe. Flow increases planned such that turbidity responses were expected to be well within regulatory limits (4 NTU limit at customers tap in the UK). These trials were executed before, to establish a base condition, just after invasive cleaning to assess the amount of material left on the pipe wall and quarterly thereafter to explore subsequent change. All flow conditioning trials were operated with repeatable conditions i.e. similar time of day, duration, locations, equipment and section of trunk main to ensure the trial results were comparable. To monitor downstream



the turbidity response, two (2) ATI NephNet turbidity loggers were used with a 1 second sampling interval to ensure data validation and confidence. Additionally a Hach handheld turbidity logger was used for on-site data verification. A specially designed ABB flow meter attached to a UK standard hydrant standpipe from Langham industrial controls was used to measure flow (and therefore shear stress) with local manual control of a gate valve.

## 3. Results and Discussions

### 3.1 Hydraulic Capacity Assessment

Hydraulic models were developed and simulated in standard EPANET software (Rossman, 2000). The hydraulic model was calibrated using monitored flow as inputs and minimising visual dissimilarities and errors between downstream simulated and measured pressure. While pipe roughness alone can produce accurate simulation of observed pressure, inaccurate representation of velocities which can be significant for quality application can persist as the above is an indeterminate problem space. Calibration optimisation processes should also consider changing pipe internal diameter particularly for large roughness heights (Boxall et al., 2004). To achieve the best fit with reduced uncertainties, the models were optimised through PEST calibration software (Doherty and Johnston, 2003). Figure 2a shows the flow, and best simulated and measured pressure data for pre invasive cleaning. As the main is around 40 year old AC, it was expected that it might have relatively high roughness due to continuous material fouling. From the optimisation, best achievable result for pre-invasive cleaning trial was 6.82 mm pipe roughness and 215 mm effective diameter.

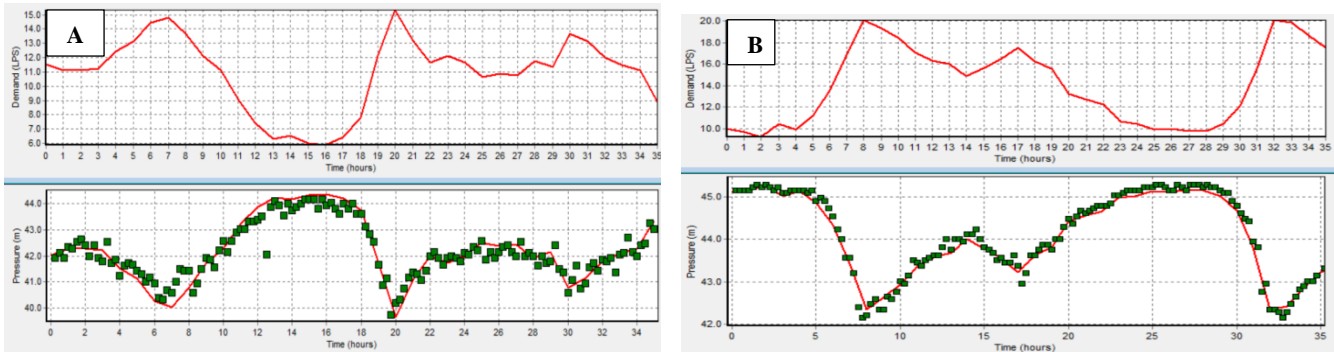

**Figure 2: Pre and post invasive cleaning hydraulic simulation in EPANET. Demand in l/s (top plot) and simulated and measured pressure head in m (bottom plot). A) pre cleaning B) post cleaning.**

After the ice-slurry cleaning process, a similar monitoring program was conducted by measuring pressure and flow. With the monitored pressure and flow data, the hydraulic model was re-optimised (figure 2b), with the best model fit achieved with a pipe roughness of 1.05 mm and effective diameter of 227mm. Thus, from hydraulic model optimisation, a seven fold reduction in roughness height was found after the invasive cleaning. This demonstrates improved mains carrying capacity and indicates that the ice slurry intervention removed material layers that cause significant losses.



A similar pressure and flow monitoring program was conducted 12 months after the invasive cleaning to quantify any further change of hydraulic capacity. The calibrated roughness was found to increase to 2.28 mm and effective diameter to reduce to 224mm. This suggests longer term accumulation, continuous particulate fouling that impact on the hydraulic capacity. Table 1 presents all the diameters and roughness values. It should be noted that the optimisation was not forced to follow the
5    relationship of Boxall et al. (2004) hypothesis, but these results do support the previous finding that 1 mm roughness produces an approximate 2 mm reduction in effective pipe diameter.

**Table 1: Hydraulic parameter optimisation results for pre-post and 12 months after invasive cleaning**

| Hydraulic Parameters and Calibration efficiency | Initial Mean Diameter | Pre Invasive Cleaning (Optimised) | Post Invasive Cleaning (Optimised) | +12 Months Post Invasive Cleaning (Optimised) |
|---|---|---|---|---|
| $k_s$ [mm] | ---- | 6.82 | 1.05 | 2.28 |
| Diameter [mm] | 228.04 | 215.00 | 227.40 | 224.04 |
| $R^2$ [-] | - | 0.9424 | 0.9681 | 0.9227 |

## 3.2 Water Quality Assessment

Water quality performance was assessed by measuring the discolouration response due to imposed excess shear stress,
10    repeated as set out in Figure 1B. Figure 3 presents daily demand and applied shear stress during the first, second and third of these increases in flow and the ice slurry pigging. Typical maximum daily shear stress before the cleaning intervention was about 1.5 N/m², estimated using calibrated roughness values. Due to operational circumstances, daily peak demand had increased after the intervention for more than a month and as a result daily shear stress was also altered for the selected trunk main. Revised maximum daily shear stress after cleaning intervention calculated from improved $k_s$ value was 1.0 N/m² and
15    later it was changed to 0.6 N/m². This value persisted throughout the following twelve months of monitoring.

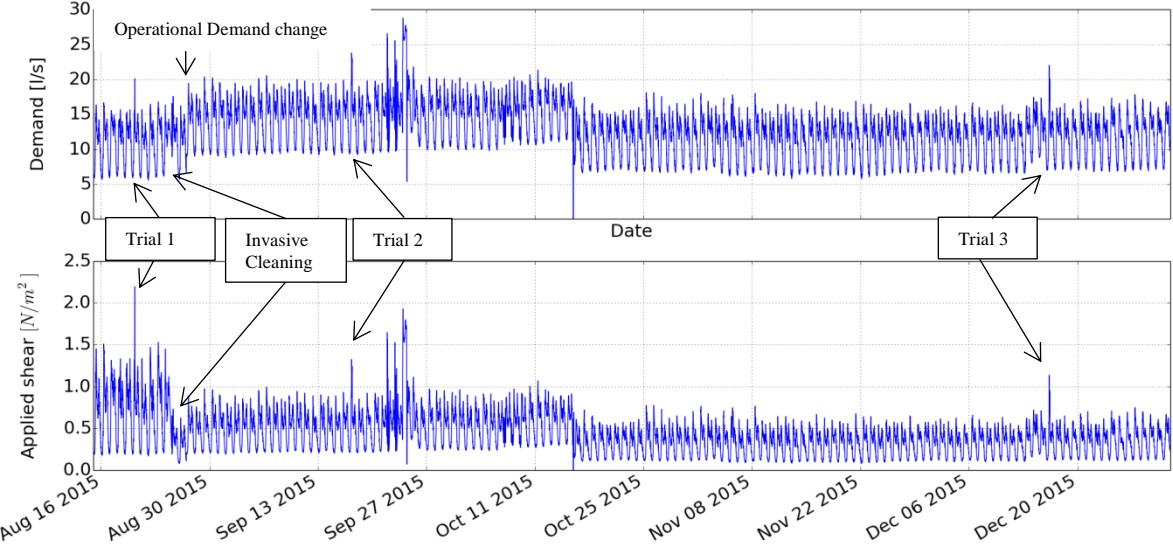

**Figure 3: Daily demand in l/s and applied shear stress in N/m² variations from August, 2016 till December, 2016**



Figure 4a shows the measured and simulated turbidity response due to shear stress increases before ice slurry pigging intervention. It is evident from figure 4a that material started to mobilize during the shear step of 1.1 to 1.65 N/m$^2$, consistent with a normal daily max shear stress of 1.5 N/m$^2$. Maximum shear stress during the trial was 2.20 N/m$^2$ generating a measured peak turbidity of about 2.0 NTU. To avoid regulatory turbidity limit (4.0 NTU), shear stress was reduced stepwise to 1.135 N/m$^2$ and it continued until the turbidity response had returned to pre-trial levels. An initial turbidity spike was observed around 8:30am due to connection and opening of the hydrant and is not considered a pipe actual turbidity response by shear stress increase.

To assess the discolouration risk from the main after it had been invasively cleaned, a similar trial was conducted 3 weeks after the pigging intervention (see trial timeline in Figure 1b). Figure 4b shows that despite the cleaning, material was mobilised from the pipe wall by only a small increase in excess shear stress. The maximum operated shear stress during the second trial was 1.32 N/m$^2$ and peak turbidity was about 1.0 NTU. As with other case studies, the ice slurry pigging was shown to remove significant amounts of materials however, these results shows that loose material was present in the pipe line only 3 weeks after the invasive cleaning.

Previous research has suggested that ice plug pressure or friction forces can drop as the trial continues (Candy et al., 2010). Therefore, potentially cleaning performance was inadequate to remove all the materials during ice plug formation and movement along pipe length. As noted previously the trial main had a low point in its longitudinal profile, as shown in figure 1a. Contractors for the invasive cleaning program suggested the ice plug may have dropped pressure while flowing upwards from the dip section, possibly leaving material there. However, figure 4b shows linear increase in material with no spike associated with travel time from the low point.

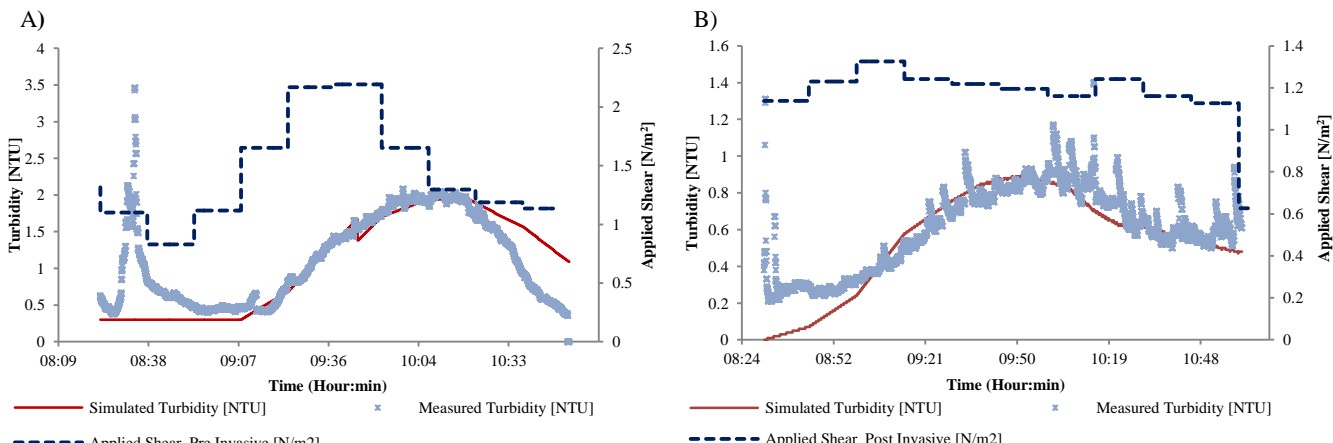

**Figure 4: Measured and simulated turbidity response A) before the invasive cleaning trial B) after the invasive cleaning trial**

Flow increases where repeated at 3 monthly intervals to assess the ongoing accumulation of loose material. Due to operational constraints the imposed shear stresses where not exactly the same, as in figure 3, hence results are not directly comparable. In order to aid comparison the time series data was converted to 'volumetric turbidity' due to specific imposed



shear increase to estimate the material release rate per unit wall area. This was calculated by integration of the time turbidity plots with respect to the amount (flow rate with time) of water used and divided by the maximum imposed shear stress and pipe internal wall area. This effectively accounts for differences in mass flux, but assumes mobilisation is linear with flow rate. While the last assumption is not valid, it is an acceptable simplification over the range of different excess shear stress

imposed here. Figure 5 summarises these amounts of volumetric turbidity mobilised from all trials, together with indicative SR outlet seasonal water temperature collected by spot samples part of regular sampling program. As expected pre-invasive cleaning trial generated relatively higher turbidity response compared to all post cleaning trials. However, the 3 weeks post invasive cleaning amount is high as well suggesting that the material was able to regenerate quicker after invasive cleaning compared to the conditioning trials, rapidly developing to re-establish equilibrium between cohesive layer forces and daily

peak shear stress, or that material loosened by the slurry was not fully removed. The following 3 months period, where temperature was lowest shows the least amount of material released, while the final 3 months period where temperature was greatest showed the overall highest levels of mobilisation of weakly adhered material. Hence this indicates a temperature influence on material regeneration process in this trunk main. Previous research evidence has also suggested a temperature impact on material regeneration processes and associated this with microbial activity with biofilms (Sharpe, 2012).

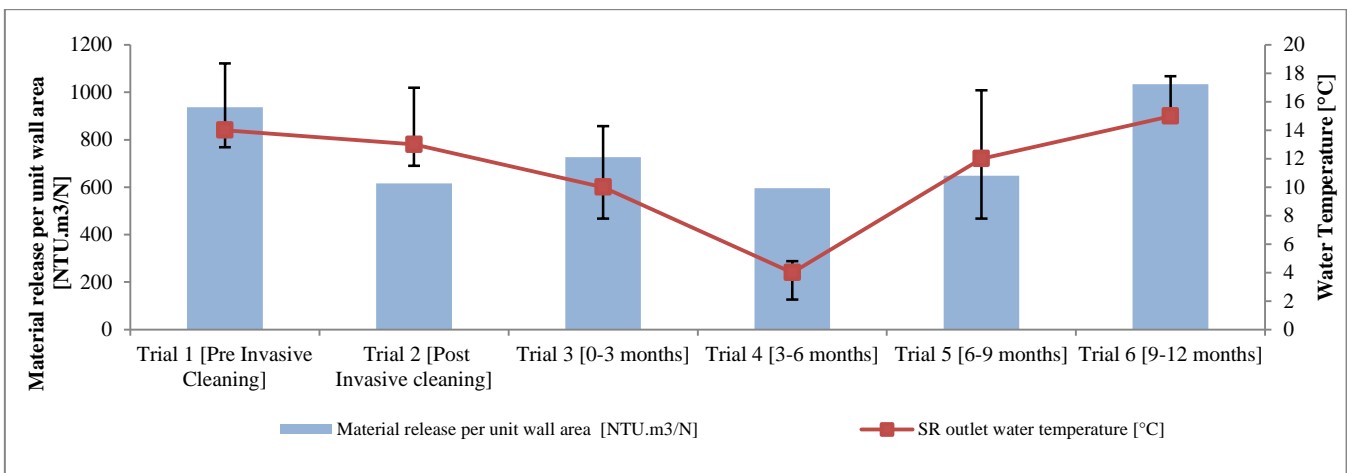


**Figure 5: Volumetric turbidity output during flow conditioning trials with service reservoir outlet temperature with seasonal variations**

### 4. Conclusions

The study investigates pre, post and long term, quality and quantity performance of a trunk main subject to invasive cleaning

by ice slurry pigging. Benefits expected due to invasive cleaning included an improvement in hydraulic capacity and a reduction in discolouration risk, as well as improve asset resilience and pipe life span. The findings from the fieldwork are summarised below:





- Hydraulic modelling and calibration showed pipe roughness reduced by about 7 times after semi-solid ice slurry cleaning intervention. However, monitoring and modelling after 12 months showed that the roughness had slowly increased after the intervention suggesting continuous material fouling impacting on the hydraulic capacity.
- Pipe wall material mobilisation through controlled shear stress increases removed significant material after the selected invasive cleaning process. This indicates that loose particles remained on the pipe wall and that there was still a risk of discolouration after this invasive cleaning intervention was performed.
- Repeated flow conditioning trials showed ongoing material accumulation, evidence of ongoing endemic processes. The amount of this fouling was observed to correlate with temperature, suggesting a biologically mediate process.

## 5. Acknowledgements

The author would gratefully thank Scottish Water for site access and data provision to complete this research work. The research work is funded by Scottish Water and Pennine Water Group, EPSRC platform grant EP/1029346/1.

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
