# Peer review of "Quantity and Quality Benefits of in-Service Invasive Cleaning of Trunk Mains"

_Drinking Water Engineering and Science, 2017_

## Referee Comment (RC1) · Anonymous Referee #1 · 20 Feb 2017

This paper deals with the impact of ice pigging on the reduction of pipe roughness, and the subsequent growth in roughness over time, from tests conducted on a real 2.4 km AC trunk main in Scotland. I commend the authors for investigating the very complex behaviour and practical difficulties associated with real water supply systems. The paper presents interesting and novel results that makes a contribution to our understanding of pipe roughness behaviour.

However, the paper needs to be improved before it can be published as described below:

1) The paper contains a number of grammatical errors and language problems. (This is surprising given the command of English that most of the co-authors have. It is frustrating as a reviewer to have to deal with issues that should not be present in

manuscripts with first-language English speakers as co-authors). Here are examples of some of the problems in the manuscript:

- Page 1, Line 15: "had be not been"

- Page 2, Line 1: The sentence has no verb.

- Page 3, Line 7: "No known cleaning intervention have ever been"

- Page 3, Line 24: "These trials were executed before, to establish a base condition,"

2) The text also contains logical errors or incomplete descriptions making it ambiguous or hard to follow (again something co-authors should have picked up on). Examples include:

- Page 2, Line 5: "Discolouration is the water quality issue most apparent to customer, causing the highest contact rates worldwide."

- Page 4, Line 8: "minimising visual dissimilarities and errors between downstream simulated and measured pressure".

- Page 4, Line 9: "While pipe roughness alone can produce accurate simulation of observed pressure". How does the pipe roughness do this?

- Page 4, Line 10: "inaccurate representation of velocities which can be significant for quality application can persist as the above is an indeterminate problem space."

- Page 4, Line 23: " Thus, from hydraulic model optimisation, a seven fold reduction in roughness height was found after the invasive cleaning." This is inaccurate. When a reduction is made, it has to be stated relative to the original value. A reduction of one fold means that the value was reduced by 100 %, i.e. to zero.

3) Nothing is mentioned in the paper on the possibility of leakage from the pipe and new leaks forming during the testing period. How would leakage have affected the results?

4) The calibrated pipe roughness values include minor losses at bends and joints. Why weren't these incorporated in the model and how will they likely impact on the results?

5) Other comments

- Acronyms should be used with discretion in publications since unfamiliar acronyms serve to obfuscate rather than clarify the text. I suggest removing 'TOTEX', 'PODDS' and 'SR'.

- Be consistent with the use of capital or small letters when referencing figures: '1A' not '1a'.

- Page 2, Line 6: What does 'international accepted' research mean? Why not simply state that research was conducted and the findings are...

- Page 2, Line 31: "mostly residential with consistent demand across the year". Do you mean that there was not seasonal variation in demand?

- Page 4, Line 1: "two (2) ATI NephNet turbidity loggers were used with a 1 second sampling interval to ensure data validation and confidence." There is no need to repeat the written 'two' with a number '2'. Using two loggers does not automatically ensure data integrity. Describe how this was done.

- Page 4, Line 13: "PEST calibration software". This software was developed for a watershed model. An explanation of the method and how it was applied to the pipe roughness problems is required.

- Page 5: An explanation of the 'operational circumstances' that lead to the changes in consumption pattern should be provided. How are these expected to have influenced the results?

- Page 6, Line 4: "To avoid regulatory turbidity limit (4.0 NTU), shear stress was reduced stepwise to 1.135 N/m2". As I understand this test, the shear values were generated by flushing the pipe through a hydrant. Why would the turbidity limit then apply, or was

consumers simultaneously connected to the system?

- Page 7, Line 7: "Benefits expected due to invasive cleaning included an improvement in hydraulic capacity and a reduction in discolouration risk, as well as improve asset resilience and pipe life span." It is not clear how invasive cleaning would improve the 'resilience' and 'life span' of a pipe.

- It will be useful to have a table with the test parameters on which Figure 5 was based to allow the reader to get a better understanding of the variations observed.

---

## Referee Comment (RC2) · T. Walski (Referee) · 22 Feb 2017

This was a very nice study. It's great to see researchers out in the field working with real water systems. I just had a few questions.

The rapid increase in turbidity in the months after the cleaning indicates that there is some source of the solids causing the turbidity. Without knowing the nature of the solids, it is difficult to determine its source. What did the solids look like—iron particles, treatment plant floc carryover, manganese solids, asbestos particles or microbial growth.

The message from this paper seems to be that if a utility cannot control the source of particles, even the best cleaning methods will only have limited success. Because the pipe was made up of asbestos cement, it is unlikely the particles responsible for

the turbidity originated in the test section. The fact that they turbidity increased after cleaning implies that the turbidity was not due to a poor job of cleaning. If the test section was the source, then sliplining or cement mortar lining would be a good way to prevent turbidity increases with time.

It would have been great to have cut a section out of the pipe and examined it. Were solids only found on the bottom of the pipe or where they uniformly distributed around the circumference? Collecting a few pipe coupons around the circumference would have been useful if a section could not be removed.

Some minor questions and observations:

What is meant by "high contact rates worldwide"? What is a contact rate?

In North America, a 228 mm pipe would not be considered a "trunk main". Depending on the system, that terminology is usually reserved for pipe on the order of 500 mm or larger.

The title referred to in-service cleaning, but it sounds as if no customers were being provide water along the test section from the test pipe during this work. Were there no customers on the line or where they provided water through bypass piping?

―――――――――――――――――――

---

## Short Comment (SC1) · 23 Feb 2017

Quantity and quality benefits of in-service invasive cleaning of trunk mains

Very nice work, definitely worth publishing.

There are some unclarities in the field work that I think you should address:

* Section 3,2 is water quality assessment, and the trials that were done are some sort of a risk assessment to see how much of a turbidity response is caused by a certain increase in shear stress (due to an increase in flow). This is a controlled flow increase test, if you like. It resembles the standardized RPM (resuspension potential method, Vreeburg and Schaap 2004), except that it was not controlled to the same increase in flow each time the test was done. I believe the term "conditioning test" should be

avoided. It is a risk assessment, not a cleaning action. Also, the PODDS explained shear stress conditioning, to avoid high future turbidity responses, is something very different. Hence, I would avoid using "conditioning" in this paper.

* It is not clear to me why the 6 trials should best be compared (in fig 5) by dividing the turbidity by the product of shear stress and pipe wall area. I would like to see fig 5 also for the clean turbidity*Q data, and a better explanation of why this division of tau is valid or could be valid.

* Why does table 1 not contain the results of trial 3, 4 and 5? Is it possible to find some sort of correlation between ks and turbidity response (corrected for shear if you like)? Could pressure and flow data indicate over time the diameter reduction and thus indicate the growth of the loose material (plus biofilm)? It would be worthwhile to check this briefly and discuss, without being able to prove this based on only one trunk main.

* If for Table 1 I add diameter and two times the roughness, I approximately get the assumed diameter of 228 mm. In the calibration test, is the sum of D and ks limited to this? If yes, please mention this. If not, would it be a good idea to do so?

* Fig 5 suggest turbidity response after 12 months was similar to pre cleaning, whereas the ks was not yet increased to the same amount.

* Fig 4b shows that PODDS was able to simulate the measured data quite well. Since the text says that the max of 1 NTU was not expected, I assume the PODDS result could not have been generated before the trial results were available. It would be helpful to clarify this in the text. I am wondering, what would PODDS have predicted based on the data of fig 4a? This could indicate what the actual results of the cleaning were. Could you use PODDS to predict for each trial what the turbidity response would be for a set controlled flow increase? Thus mimicking the test under the same conditions, and then compare the results. In which case the division by shear stress would not be needed.

* Fig 3: shear stress during ice pigging must be much higher, but is not easy to calculate. Instead, I would leave this part out to avoid confusion. What happened around 27 September (downstream pipe break?)? Caption should say 2015.

* I do not understand how asset deterioration (for other than cast iron pipes) would lead to water quality issues. I can see that if no cleaning is done, time will cause more particulate accumulation, but this does not relate to the age or the condition of the pipe.

* This is one of several studies that "suggests" a temperature dependence. The reference to Sharpe's thesis is very limited. The biofilm explanation is not substantiated with this particular AC trunk main study.

There are quite some grammar mistakes and typos that need to be looked at. For a conference paper the limited number of references was ok, but I would like to see an introduction with some more references added, in order to place this work more in perspective.

―――――――――――――――――――――

---

## Short Comment (SC2) · 28 Feb 2017

I have reviewed the paper, and have the following comments: 1. page 3 line 20, "...shear stress would induced...", should it be "...shear stress would induce..."? 2. In Section 3.1, authors mentioned that the hydraulic model calibration was conducted by adjusting both pipe diameter and roughness height, which may very likely compensate each other to minimize the difference between the simulated and observed values. Authors should elaborate on how the compensation error be avoided to achieve the relatively accurate estimate of pipe diameter and roughness height.

---

## Author Comment (AC1) · 21 Apr 2017

**Referee 1: Anonymous**

First of all, the authors would like to thank the reviewer for taking valuable time to review and for the critical assessment of the paper.

1) **Comment 1:** Grammatical and language issue

C1 Ans: We will correct the stated grammatical errors and language problems the reviewer stated in the comments sections.

**Comment 2 :** The text also contains logical errors or incomplete descriptions making it ambiguous or hard to follow (again something co-authors should have picked up on). Examples include:

• Page 2, Line 5: "Discolouration is the water quality issue most apparent to customer, causing the highest contact rates worldwide."

Ans: We are not sure what is unclear in this sentence.

Regarding contact rate term, we think the rate term is correct i.e. number of customer contacts per 1000 population per year. Otherwise, the contact numbers are not comparable.

• Page 4, Line 8: "minimising visual dissimilarities and errors between downstream simulated and measured pressure"

Ans: Wording of the sentence has been ammended.

"minimising visual dissimilarities and maximising correlation coefficient ($R^2$) between downstream simulated and measured pressure."

• Page 4, Line 9: "While pipe roughness alone can produce accurate simulation of observed pressure". How does the pipe roughness do this? :

Ans: To conduct a hydraulic modelling calibration, it is a standard practice to change only pipe roughness to reduce the difference between measured and simulated pressure – i.e. increase roughness until headloss is sufficient to match the observed pressures. However, this can result in unrealistically large roughness values and erroneous velocities and travel times that are particularly important for water quality simulation. Boxall et al., (2004)[1] showed that 1 mm pipe roughness value effectively reduces pipe diameter by 2 mm, matching both pressure and travel time data, suggesting the importance of changing both pipe roughness and diameter simultaneously.

• Page 4, Line 10: "inaccurate representation of velocities which can be significant for quality application can persist as the above is an indeterminate problem space.":

Ans: This sentence refers to the necessity to simulate velocities and hence travel times for water quality accurately. For a given imposed flow, various combinations of diameter and roughness can produce similar headloss and pressure – an indeterminate problem space. Each of these paired values has a unique velocity. It is important that the correct pairing is selected to simulate water quality effects.

The full sentence we agree however is unwieldy so has been edited to "While pipe roughness ($k_s$) alone can be modified to produce an accurate simulation of observed pressure, inaccurate representation of velocities, which can be significant for the quality applications, can persist. This is because hydraulic calibration is an indeterminate problem space where various combinations of diameter and roughness can produce similar headloss and pressure.

• Page 4, Line 23: "Thus, from hydraulic model optimisation, a seven fold reduction in roughness height was found after the invasive cleaning." This is inaccurate. When a reduction is made, it has to be stated relative to the original value. A reduction of one fold means that the value was reduced by 100 %, i.e. to zero:

Ans: We think it is correct. The same explanation is given to the Oxford dictionaries "https://en.oxforddictionaries.com/definition/sevenfold".

2) **Comment 3:** Nothing is mentioned in the paper on the possibility of leakage from the pipe and new leaks forming during the testing period. How would leakage have affected the results?

C3 Ans: The following has been added to the Results and discussion chapter:

The trunk main studied had no known leakage, as assessed through night line analysis. The effect of any unknown background leakage would have been manifest in the flow data that was used as an input to the model. The night line was not observed to change from the start to the end of the monitoring period (other than due to known operational changes) suggesting no new leakage occurred during the study.

3) **Comment 4:** The calibrated pipe roughness values include minor losses at bends and joints. Why weren't these incorporated in the model and how will they likely impact on the results?

C4 Ans: We had developed the hydraulic model as realistically as possible from industry records and local operation knowledge. During the model construction, minor losses were incorporated as EPANET loss coefficient inputs, determined from the EPANET manual Table 3.3 (p-32)[2]. A comment to explain this has been added to the manuscript. These values were not considered as calibration variables, and where fixed throughout the simulations so effects would have been constant.

4) **Comment 5**

• Acronyms should be used with discretion in publications since unfamiliar acronyms serve to obfuscate rather than clarify the text. I suggest removing 'TOTEX', 'PODDS' and 'SR'

Ans: PODDS (Prediction of Discolouration in Distribution Systems) term has been using in the academic literature since 2001. The PODDS model theory drives the discolouration risks assessment for this case study. A change of SR to 'service reservoir' and TOTEX to 'total expenditure' has been added to the manuscript.

• Be consistent with the use of capital or small letters when referencing figures: '1A' not '1a'.

Ans: All figure referencing has been changes to small letters in the manuscript i.e. 1a

• Page 2, Line 6: What does 'international accepted' research mean? Why not simply state that research was conducted and the findings are...

Ans: The PODDS model has published simulated discolouration responses from the UK and other countries e.g. Australia (Boxall and Prince, 2006)[3], Portugal (Husband and Boxall, 2016)[4] etc. So we think this is conceptually correct to include the international status, however agree 'accepted' is not appropriate and we have changed this to 'validated'.

• Page 2, Line 31: "mostly residential with consistent demand across the year". Do you mean that there was not seasonal variation in demand?

Ans: Yes.

• Page 4, Line 1: "two (2) ATI NephNet turbidity loggers were used with a 1 second sampling interval to ensure data validation and confidence." There is no need to repeat the written 'two' with a number '2'. Using two loggers does not automatically ensure data integrity. Describe how this was done.

Ans: (2) has been removed from the manuscript.

The below explanation has been added to the paper:
The ATI turbidity loggers were calibrated under laboratory conditions and using two loggers to ensure that the collected data was consistent. The spot check of these instrument outputs was tested via HACH handheld logger which was calibrated against formazin turbidity standard samples.

• PEST calibration software". This software was developed for a watershed model. An explanation of the method and how it was applied to the pipe roughness problems is required:

Ans: PEST is a model independent calibration software (Doherty, 2005)[5] and has been extensively tested for various watershed models. PEST calibration ability was also tested for EPANET model calibration (Méndez et al., 2013)[6]. The model was previously integrated into

MODFLOW, a groundwater modelling software as well. These new references have been added to the manuscript appropriate section.

The below explanation of the method has been added to Chapter (3.1):
The estimated boundary condition of pipe roughness and the diameter using Boxall et al. (2004)[1] concept has been applied to the PEST in conjunction with the EPANET model to determine the best possible solutions comparing simulated and measured downstream pressure.

- Page 5: An explanation of the 'operational circumstances' that lead to the changes in consumption pattern should be provided. How are these expected to have influenced the results?

Ans: An explanation of the operational circumstances reasons has been added to the discussion chapter (3.2):
From 27 August 2015 till 16 October 2015, a few new properties were connected to the downstream distribution zone fed from the investigated trunk main during additional repair work in a neighbouring network. Demand was increased by about 3 l/s during this process which can be confirmed from the continuous night line profile for over two months.

- Page 6, Line 4: "To avoid regulatory turbidity limit (4.0 NTU), shear stress was reduced stepwise to 1.135 N/m$^2$". As I understand this test, the shear values were generated by flushing the pipe through a hydrant. Why would the turbidity limit then apply, or was consumers simultaneously connected to the system?:

Ans: The reviewer is correct that these operations were undertaken by opening fire hydrants; however, this was only to achieve additional flow and associated shear stress. The majority of the flow was due to the downstream demands of the associated network, and hence the regulatory limit applied.

- Page 7, Line 7: "Benefits expected due to invasive cleaning included an improvement in hydraulic capacity and a reduction in discolouration risk, as well as improve asset resilience and pipe life span." It is not clear how invasive cleaning would improve the 'resilience' and 'life span' of a pipe:

Ans: Resilience encompasses many factors, including hydraulic capacity which was clearly improved here. Life span has been removed.

- It will be useful to have a table with the test parameters on which Figure 5 was based to allow the reader to get a better understanding of the variations observed:

Ans: An explanation of material release rate and accounting variables was in the manuscript Page 6: line 22-25 and Page 7: line 1-5. Addition to this, a table of test parameters has been added to the manuscript:

*Table 1: Test parameters for material release rate calculations*

| Parameters | Unit | Trial 1 | Trial 2 | Trial 3 | Trial 4 | Trial 5 | Trial 6 |
|---|---|---|---|---|---|---|---|
| Diameter, D | m | 0.215 | 0.2274 | 0.2274 | 0.2274 | 0.2274 | 0.22404 |
| $K_s$ | mm | 6.82 | 1.05 | 1.05 | 1.05 | 1.05 | 2.28 |

Though the $k_s$ and diameter values were unknown in trial 3, 4 and 5, paired values ($k_s$ and D) were assumed to be equal to trial 2.

To calculate imposed excess shear stress addition to the above table, discolouration material density ($\rho$) and gravity (g) was used at 1100 Kg/m$^3$ (Boxall et al., 2001; Ryan et al., 2008)[7,8] and 9.81 m/s$^2$.

References stated in this discussion papers:

1.  J. B. Boxall, A. J. Saul, and P. J. Skipworth, 'Modeling for Hydraulic Capacity', Journal - American Water Works Association, vol. 96, no. 4, pp. 161–169, Apr. 2004.

2.  L. . Rossman, 'Epanet 2: User Manual', USEPA, Cincinnati, OH, User Manual EPA/600/R-00/057, Sep. 2000.

3.  J. Boxall and R. Prince, 'Modelling discolouration in a Melbourne (Australia) potable water distribution system.', J Water SRT - Aqua, vol. 55, pp. 207–219, 2006.

4. S. Husband and J. Boxall, 'Understanding and managing discolouration risk in trunk mains', Water Research, vol. 107, pp. 127–140, Dec. 2016.

5. J. Doherty, 'PEST - Model-Independent Parameter Estimation', Watermark Numerical Computing, Brisbane, Australia, User Manual, 2005.

6. M. Méndez, J. A. Araya, and L. D. Sánchez, 'Automated parameter optimization of a water distribution system', *Journal of Hydroinformatics*, vol. 15, no. 1, pp. 71–85, Jan. 2013.

7. J. Boxall, P. J. Skipworth, and A. Saul, 'A novel approach to modelling sediment movement in distribution mains based on particle characteristics', in *ater Software Systems: v. 1: Theory and Applications (Water Engineering & Management)*, Hertfordshire, UK: Research Studies Press, 2001, pp. 263–273.

8. G. Ryan *et al.*, 'Particles in Water Distribution System: Characteristics of particulates Matter in Drinking Water Supplies', CRC, Australia, Researh Report 33, 2008.

---

## Author Comment (AC2) · 21 Apr 2017

**Referee 2: Tom Walski**

First of all, the authors would like to thank the reviewer for taking valuable time to review and for the critical assessment of the paper.

**Comment1**: It would have been great to have cut a section out of the pipe and examined it. Were solids only found on the bottom of the pipe or where they uniformly distributed around the circumference? Collecting a few pipe coupons around the circumference would have been useful if a section could not be removed.

**C1 Ans**: A pre-intervention pipe cut out was taken from the trunk mains low point (Figure 1a in the manuscript) to assess the pipe internal condition and amount of accumulated material present on the asbestos cement pipe wall.

The following information has been added to the paper:

Figure 1 shows images of pre-interventon pipe cut out. Accumulated material can clearly be seen around the full pipe circumference, supporting the PODDS model concepts. The cut out was taken at the longitudinal low point (manuscript Figure 1a), such that all gravitationally driven self-weight settling processes that would have led to invert deposits were explored, with none being found.

[Figure]

[Figure]

*Figure 1: Pre-cleaning intervention pipe cut out*

**Comment 2:** The rapid increase in turbidity in the months after the cleaning indicates that there is some source of the solids causing the turbidity. Without knowing the nature of the solids, it is difficult to determine its source. What did the solids look likeǎAˇTiron particles, treatment plant floc carryover, manganese solids, asbestos particles or microbial growth.

**C2 Ans:** The following has been added to the discussion, it should be noted that conclusion cannot be drawn based on the data collected.

Results of metal samples during trials are shown in figure 2. Figure 2a shows all metal samples during trials where the concentration of manganese (Mn) was high and occasionally exceeding the UK regulatory prescribed concentration value (PCV) value of 50 $\mu$g/l. Iron and aluminium concentrations in the bulk water is also shown to be significant, although well below the UK PCV limit. Figure 2b presents the results for calculated metal concentrations for an equivalent 1.0 NTU limit. Manganese PCV is likely to be exceeded during all trials at this threshold suggesting high Mn content in the bulk water. From this it could be suggested that the accumulation or fouling effects are driven by manganese and other metal (e.g. iron and aluminium) precipitation from the bulk water which is consistent with previous research findings (Boxall et al., 2003; Husband and Boxall, 2011; Seth et al., 2004)[1,2,3]. However, a complete conclusion about inorganic particles responsible here for discolouration risks cannot be drawn from this sampling study alone as undisturbed sampling data from trunk main were unavailable. Also the previous work has indicated that biological processes impact on material accumulation and hence it's influence discolouration risks as well (Gauthier et al., 1999; Husband et al., 2016)[4,5].

[Figure]

*Figure 2: a) Metal concentration in bulk water during trial durations, b) metal concentration equivalence at 1.0 NTU*

**Comment 3:** "High contacts rate worldwide?" What is a contact rate?:

**C3 Ans:** We think the rate term is correct i.e. number of customer contacts per 1000 population per year. Otherwise, the contact numbers cannot be comparable.

**Comment 4:** In North America, a 228 mm pipe would not be considered a "trunk main". Depending on the system, that terminology is usually reserved for the pipe on the order of 500 mm or larger.

**C4 Ans:** A transmission (trunk) main is defined by its operation, not its size. A trunk main is one that is used to transport water between treatment works, service reservoirs, demand zone etc. Typically it does not have customers directly connected to it (with the occasional unavoidable expectations). This definition has been added to the paper.

**Comment 5:** The title referred to in-service cleaning, but it sounds as if no customers were being provide water along the test section from the test pipe during this work. Were there no customers on the line or where they provided water through bypass piping?

**C5 Ans:** As commented above (**C4**) this was a trunk main so did not have any direct customer connections, however the downstream network and associated consumers were connected and operational throughout.

**Reference cited in this discussion paper:**

[1]     J. Boxall, P. J. Skipworth, and A. J. Saul, 'Aggressive flushing for discolouration event mitigation in water distribution networks', *Water Science & Technology: Water Supply*, vol. 3, no. 1–2, pp. 179–186, 2003.

[2]     P. S. Husband and J. B. Boxall, 'Asset deterioration and discolouration in water distribution systems', *Water Research*, vol. 45, no. 1, pp. 113–124, Jan. 2011.

[3]     A. Seth, R. Bachmann, J. Boxall, A. Saul, and R. Edyvean, 'Characterisation of materials causing discolouration in potable water systems', *Water Sci. Technol.*, vol. 49, no. 2, pp. 27–32, 2004.

[4]     S. Husband, K. E. Fish, I. Douterelo, and J. Boxall, 'Linking discolouration modelling and biofilm behaviour within drinking water distribution systems', *Water Science and Technology: Water Supply*, vol. 16, no. 2, p. ws2016045, Apr. 2016.

[5]     V. Gauthier, B. Gérard, J.-M. Portal, J.-C. Block, and D. Gatel, 'Organic matter as loose deposits in a drinking water distribution system', *Water Research*, vol. 33, no. 4, pp. 1014–1026, Mar. 1999.

---

## Author Comment (AC3) · 21 Apr 2017

**Referee 3: Mirjam Blokker**

First of all, the authors would like to thank the reviewer for taking valuable time to review and for the critical assessment of the paper.

**Comment 1:** Section 3,2 is water quality assessment, and the trials that were done are some sort of a risk assessment to see how much of a turbidity response is caused by a certain increase in shear stress (due to an increase in flow). This is a controlled flow increase test, if you like. It resembles the standardized RPM (resuspension potential method, Vreeburg and Schaap 2004), except that it was not controlled to the same increase in flow each time the test was done. I believe the term "conditioning test" should be avoided. It is a risk assessment, not a cleaning action. Also, the PODDS explained shear stress conditioning, to avoid high future turbidity responses, is something very different. Hence, I would avoid using "conditioning" in this paper.

**C1 Ans:** The distinction between a risk assessment and cleaning action is interesting. The network intervention of the RPM is a cleaning action – removing material from the network due to increase in hydraulic forces. How the data is then interpreted can be a part of a risk assessment. Interpretation of the RPM may be complicated since it uses a fixed velocity criterion, which then exerts a diameter dependent force and also takes no cognisance of normal, daily or recent hydraulic conditions. To overcome this, we used an excess shear stress criterion. However, due to the invasive cleaning application, both pipe roughness ($k_s$) and diameter changed to such an extent that the applied excess shear could not be systematically achieved.

Any increase in hydraulic forces above those normal experiences will remove material and hence have a cleaning action. When such an increase is small in magnitude and duration, it is a long way from removing all material and hence has a conditioning effect – removing all material with adhered strength up to the imposed force. This is exactly consistent with the PODDS based shear stress conditioning concept.

**Comment 2:** It is not clear to me why the 6 trials should best be compared (in fig 5) by dividing the turbidity by the product of shear stress and pipe wall area. I would like to see fig 5 also for the clean turbidity*Q data, and a better explanation of why this division of tau is valid or could be valid.

**C2 Ans:** The volumetric turbidity calculation by multiplying the volume of water and measured turbidity gives the mass flux effect for a specified period. Based on the PODDS concept, material layers are held in various strength profiles adhered on the complete pipe surface. This volume will be a complex function of the surface affected area and the imposed shear force. Hence when pipe area and imposed force are not constant, normalisation is required. Arguably the change in the area here was small and had little effect, but was included for completeness. The invasive cleaning produced a substantial change in roughness and hence imposed excess shear stress. Therefore, it was vital that the results were normalised by this in order to be comparable.

**Comment 3:** Why does table 1 not contain the results of trial 3, 4 and 5? Is it possible to find some sort of correlation between ks and turbidity response (corrected for shear if you like)? Could pressure and flow data indicate over time the diameter reduction and thus indicate the growth of the loose material (plus biofilm)? It would be worthwhile to check this briefly and discuss, without being able to prove this based on only one trunk main.

**C3 Ans:** Unfortunately, practical constraints where such that the data necessary for detailed hydraulic calibration was not collected for all events.

Previous work (Boxall et al., 2003)[1] has suggested that the change in roughness (~0.01mm) corresponding to notable turbidity (~ 10NTU) response for a 3inch 1.6km cast iron pipe is significantly less than the accuracy of the hydraulic calibration possible here. Hence while the above is a very worth concept and one we would wish to explore, however, it was not feasible here.

**Comment 4:** If for Table 1 I add diameter and two times the roughness, I approximately get the assumed diameter of 228 mm. In the calibration test, is the sum of D and ks limited to this? If yes, please mention this. If not, would it be a good idea to do so?

**C4 Ans:** As with the original work (Boxall et al. 2004)[2], the paired value (roughness and diameter) solution is constrained by the original pipe diameter. Comment to this effect has been added to the paper.

**Comment 5:** Fig 5 suggest turbidity response after 12 months was similar to pre cleaning, whereas the ks was not yet increased to the same amount.

**C5 Ans:** This result is covered in the manuscript, that the invasive cleaning was effective in removing significant amounts of historical accumulations. However, the remaining material was not necessarily mobilised easily through invasive cleaning exercise and hence represent a discolouration risk. After 12 months this weaker material risk had returned, but the stronger material apparently responsible for the initial roughness height had not.

**Comment 6:** Fig 4b shows that PODDS was able to simulate the measured data quite well. Since the text says that the max of 1 NTU was not expected, I assume the PODDS result could not have been generated before the trial results were available. It would be helpful to clarify this in the text. I am wondering, what would PODDS have predicted based on the data of fig 4a? This could indicate what the actual results of the cleaning were. Could you use PODDS to predict for each trial what the turbidity response would be for a set controlled flow increase? Thus mimicking the test under the same conditions, and then compare the results. In which case the division by shear stress would not be needed.

**C6 Ans:** We are not sure where the text says max 1.0 NTU was not expected after invasive cleaning. It was assumed that there could be low response due to the pipe wall cleaned with the invasive application. Did you mean that the response of 1.0 NTU turbidity target of conditioning trial?

This is an interesting question regarding mimicking the response. PODDS is a semi-empirical model with model parameters requiring calibration, although previously calibrated parameters have shown to be transferable. Before the simulation, we tested the model response using previously calibrated parameters and created a scenario profile. However, depending on the network conditions, the measured response can be varied e.g. flow estimation at service reservoir outlet, flow fluctuation of the standpipe. In practice, the target turbidity response of 1.0 NTU for flow conditioning trial has a certain buffer for not exceeding the threshold limit. During the trial (figure 4a), the response was recorded up to 2.0 NTU, although as long as the response was below the regulator limit of 4.0 NTU it was OK.

Mimicking was not effectively possible from pre to post cleaning as the PODDS model cannot track simultaneously accumulated layer mobilisation-accumulation processes. It is possible to simulate the discolouration response for a set of flow increase from trial 3 to 6 taking information from figure 4b model parameter. This is our next step to evaluate the quality performance so that we can avoid extensive fieldwork. However, it is still good to have some degree of fieldwork information so that we can accept and validate the performance assessment. The mimicking is effectively possible with the newly developed Variable Conditioning Discolouration (VCD) model by Furnass, (2015)[3] which can track both mobilisation and regeneration for a long term. However, to simulate this process in VCD model, we do require both continuous flow, upstream-downstream turbidity response which is unavailable for this study.

**Comment 7:** Fig 3: shear stress during ice pigging must be much higher, but is not easy to calculate. Instead, I would leave this part out to avoid confusion. What happened around 27 September (downstream pipe break?)? Caption should say 2015.

**C7 Ans:** We agree with the reviewer that estimation of shear stress due to the ice plug formation is hard, if not impossible.

A planned night time downstream network flushing was undertaken from 22nd September till 23rd September (2 nights). Also, there was a downstream burst event recorded as starting on 24th September and continuing until 25th September (figure 1). Unfortunately, we did not have any turbidity loggers deployed on this trunk main which could have captured the mobilisation response due to the burst event. These events could have mobilised accumulated material as it was higher than trial 2 event and potentially influence trial 3 responses. These event details have been added to the paper.

[Figure]

*Figure 1: Flow and shear stress data from referee comment section*

**Comment 8:** I do not understand how asset deterioration (for other than cast iron pipes) would lead to water quality issues. I can see that if no cleaning is done, time will cause more particulate accumulation, but this does not relate to the age or the condition of the pipe.

**C8 Ans**: We have used deterioration in a broad sense, i.e. to encompass material accumulation at the pipe wall, whether this is from the bulk water in all pipe types or to also include corrosion processes within cast iron pipes. The amount of material accumulated at the pipe wall is a prime indicator of the asset condition.

**Comment 9:** This is one of several studies that "suggests" a temperature dependence. The reference to Sharpe's thesis is very limited. The biofilm explanation is not substantiated with this particular AC trunk main study.

**C9 Ans**: We agree that this work contributes to the body of research that suggests temperature dependence, but does not definitively prove it. Sharpe's thesis is a very relevant work rigorously proving this link, and that it is biofilm-dependent, further publications from this work are not yet in print so cannot be referenced. Further references that suggest temperature dependence have been added in the paper including e.g. Blokker and Schaap, (2015)[4]. Regarding AC main biofilms, some published papers are stating how pipe material influence biofilms structure. There is no reason or evidence in previous research to suggest biofilms within AC pipe would be fundamentally different to other pipes. Hence the suggestion made is reasonable. However, this is a single study and hence not possible to confirm anything without investigating similar test profile on varying pipe material.

**Comment 10**: There are quite some grammar mistakes and typos that need to be looked at. For a conference paper the limited number of references was ok, but I would like to see an introduction with some more references added, in order to place this work more in perspective.

**C10 Ans**:

- We will correct the grammar and typing mistakes.

- The literature review and references have been revisited and updated in response to the reviewer comment; however, the changes are limited by the strict paper length constraints.

References stated in this discussion papers:

1. J. B. Boxall, A. J. Saul, and P. J. Skipworth, 'Modeling for Hydraulic Capacity', *Journal - American Water Works Association*, vol. 96, no. 4, pp. 161–169, Apr. 2004.

2. J. Boxall, A. Saul, J. D. Gunstead, and N. Dewis, 'Regeneration of Discolouration in Distribution Systems', in *World Water & Environmental Resources Congress 2003*, Philadelphia, Pennsylvania, United States, 2003, pp. 1–9.

3. W. R. Furnass, 'Modelling both the continual accumulation and erosion of discolouration material in drinking water distribution systems', Doctorate of Philosophy, University of Sheffield, Sheffield, 2015.

4. E. J. M. Blokker and P. G. Schaap, 'Temperature Influences Discolouration risk', in Computing and Control for the Water Industry (CCWI), United Kingdom, 2015, vol. 119, pp. 280–289.

---

## Author Comment (AC4) · 21 Apr 2017

**Short Review: Zheng Hu**

First of all, the authors thank the reviewer to take valuable time to review and for the critical assessment of the paper.

**Comment1:** I have reviewed the paper, and have the following comments: 1. page 3 line 20,"...shear stress would induced...", should it be "...shear stress would induce..."? 2. In Section 3.1, authors mentioned that the hydraulic model calibration was conducted by adjusting both pipe diameter and roughness height, which may very likely compensate each other to minimize the difference between the simulated and observed values. Authors should elaborate on how the compensation error be avoided to achieve the relatively accurate estimate of pipe diameter and roughness height.

**C1 Ans:**

We will correct "induced".

The diameter and roughness calibration followed the finding of Boxall et al. (2004)* that a 1mm increase in roughness reduces the diameter by 2mm and provides a unique solution to both head loss and velocity/travel time. Travel time was also confirmed here with reference to the measured and simulated turbidity (manuscript figure 4).

* J. B. Boxall, A. J. Saul, and P. J. Skipworth, 'Modeling for Hydraulic Capacity', *Journal - American Water Works Association*, vol. 96, no. 4, pp. 161–169, Apr. 2004.